# Physiological and Genetic Dissection of Sucrose Inputs to the *Arabidopsis thaliana* Circadian System

**DOI:** 10.3390/genes10050334

**Published:** 2019-05-02

**Authors:** Koumis Philippou, James Ronald, Alfredo Sánchez-Villarreal, Amanda M. Davis, Seth J. Davis

**Affiliations:** 1Department of Plant Developmental Biology, Max-Planck Institute for Plant Breeding Research, D50829 Cologne, Germany; iakk2003@yahoo.gr (K.P.); asanchezv@colpos.mx (A.S.-V.); amanda.davis@york.ac.uk (A.M.D.); 2Department of Biology, University of York, York YO10 5DD, UK; jar563@york.ac.uk; 3Colegio de Postgraduados campus Campeche, Campeche CP 24450, Mexico

**Keywords:** circadian rhythm, sucrose, CCA1, LHY, pathway

## Abstract

Circadian rhythms allow an organism to synchronize internal physiological responses to the external environment. Perception of external signals such as light and temperature are critical in the entrainment of the oscillator. However, sugar can also act as an entraining signal. In this work, we have confirmed that sucrose accelerates the circadian period, but this observed effect is dependent on the reporter gene used. This observed response was dependent on sucrose being available during free-running conditions. If sucrose was applied during entrainment, the circadian period was only temporally accelerated, if any effect was observed at all. We also found that sucrose acts to stabilize the robustness of the circadian period under red light or blue light, in addition to its previously described role in stabilizing the robustness of rhythms in the dark. Finally, we also found that CCA1 is required for both a short- and long-term response of the circadian oscillator to sucrose, while LHY acts to attenuate the effects of sucrose on circadian period. Together, this work highlights new pathways for how sucrose could be signaling to the oscillator and reveals further functional separation of CCA1 and LHY.

## 1. Introduction

Circadian clocks are endogenous timekeepers that coordinate internal physiological responses to the predicted external environment. In Arabidopsis, 30% of the transcriptome is circadian regulated, and this includes processes such as metabolism, the induction of flowering, growth, responsiveness to hormones, and biotic and abiotic stress [1,2,3,4,5,6,7,8]. Consequently, having an internal oscillator that closely matches external time enhances plant fitness [9]. Studies that investigated responsiveness to periodic stress cues or metabolism have highlighted the importance of metabolic oscillations in the regulation of circadian rhythms [3,10]. It has been proposed that endogenous timekeepers not only predict external stress cues but also provide the basis for temporal segregation between incompatible cellular metabolic processes that would otherwise be energetically futile and stressful [7,8,11,12].

Plant circadian rhythms are generated through a series of transcriptional-translational loops. At the center of the oscillator is a repressive feedback loop formed between the morning expressed MYB transcription factors CIRCADIAN CLOCK ASSOCIATED 1 (CCA1) and LATE ELONGATED HYPOCOTYL (LHY) and the night-phased TIMING OF CAB EXPRESSION 1 (TOC1), also known as PSEUDO RESPONSE REGULATOR 1 (PRR1) [13,14,15]. This core loop is subsequently regulated by a series of morning and evening loops. In the morning, sequential expression of PRR9/7/5 starting with PRR9 just after dawn results in the repression of CCA1/LHY expression throughout the day and early evening [16]. At dusk, GIGANTEA (GI) and ZEITLUPPE (ZTL) co-associate to degrade TOC1 [17,18], and the evening complex, composed of EARLY FLOWERING3, EARLY FLOWERING4, and LUX ARRYTHMO (LUX), represses the expression of *GI*, *LUX*, *PRR9*, and *PRR7* [19].

The entrainment of the oscillator occurs through perception of external cues such as light and temperature [20,21,22] but also through internal cues signals such as sucrose availability [8,23]. It has been previously shown that sucrose application shortens the circadian period, but this effect is dependent on the light level used [10,24,25]. Conversely, inhibition of photosynthesis either through growth in a CO_2_-free environment or by inhibiting photosystem II activity through 3-(3,4-dichlorophenyl)-1, 1-dimethylurea (DCMU) treatment results in lengthening of the free-running period [10].

Some of the mechanisms through which sucrose regulates the oscillator have been recently uncovered. In response to low sugar levels, the bZIP63 transcription factor directly promotes the expression of *PRR7* to promote the repression of *CCA1* [10,26]. bZIP63 activity is subsequently regulated by the sugar-sensing kinase SUCROSE NON-FERMENTING RELATED KINASE 1 (SnRK1) [27]. In the absence of any one of these components, the oscillator becomes insensitive to pulses of sucrose, highlighting the importance of these factors in sucrose-mediated entrainment of the oscillator [26]. The perception of sucrose availability may occur through the perception of the sugar-signaling molecule TREHALOSE 6-PHOSPHATE. Mutations within the Tre6P synthesis enzyme TREHALOSE-6-PHSOPHATE SYNTHASE1 (TSP1) result in insensitivity of the oscillator to sucrose [26]. Alongside this pathway, AKIN10, a sub-unit of SNRK1, also regulates the phase of *GI* expression through interactions with TIME FOR COFFEE (TIC) [28,29,30,31]. Additionally, PHYTCHROME INTERACTING FACTORS (PIFs) controls the expression of *CCA1/LHY* in a sucrose-mediated entrainment pathway [32]. This pathway is dependent on light and temperature, highlighting the intersection of external and internal cues in providing robust oscillations [33]. Separately to entrainment, sucrose also controls the amplitude and stability of circadian rhythms by stabilizing GI [25,34]. Sucrose therefore has multiple input pathways to regulate not only the period and phase of the oscillator but also its stability.

To examine how sucrose is signaling to the oscillator, here multiple luciferase reporter genes were analyzed with or without sucrose under free-running conditions. Our work confirms previous work that found sucrose does shorten the circadian period, and this process is dependent on sucrose being available under free-running conditions. Moreover, we found that the oscillator is highly responsive to sucrose availability, and shortening of the free-running period is observed with concentrations of sucrose as low as 0.5% w/v. We also observed that sucrose stabilizes circadian oscillations under free-running conditions in the light, complementing the previously observed role of sucrose in supporting circadian rhythms in the dark. Finally, we identified a role for *GI* in a blue-light-dependent sucrose signaling pathway, and we also identified noncomplementary roles of CCA1 and LHY in sucrose-mediated regulation of the oscillator.

## 2. Methods

### 2.1. Plant Material

Circadian rhythmicity was monitored using the promoter::luciferase system (Southern and Millar, 2005). The Col-0 *GI::LUC* and Ws-2 *CAB2::LUC* lines have been previously described [35,36,37]. The *cca1-11*, *lhy-21*, and *cca1-11/lhy-21 CAB2::LUC* mutants have also been described previously [36]. *CCA1::LUC* and *GI::LUC* were introgressed into the *gi-11* mutant [38] through crossing. *gi-11 CCA1::LUC* and *gi-11 GI::LUC* were then self-fertilized until stable lines were obtained.

### 2.2. Growth Conditions and Luciferase Imaging

Seeds were surface-sterilized and sown onto MS medium containing 3% w/v sucrose or no sucrose as indicated, and then stratified at 4 °C for 3 days. Seedlings were entrained for seven days under 12 h light/12 h dark photoperiods (white light at 100 µmol m^−2^ s^−1^) and a constant temperature of 22 °C. On day six, seedlings were transferred to 96-well imaging microtiter plates (Perkin Elmer, Juegesheim, Germany) containing 3% w/v sucrose or no sucrose as indicated and re-entrained for a further day before being transferred to the TOPCOUNT (Perkin-Elmer (Perkin-Elmer-Cetus), Norwalk, CT, USA). TOPCOUNT experiments were carried out under a constant temperature of 21 °C and either constant blue or constant red light as stated in text. Data were analyzed as previously described [20,21,39].

## 3. Results

### 3.1. Confirmations That Sucrose Regulates Free-Running Period of the Oscillator

To explore the effects of sucrose on the oscillator, seedlings expressing *GI::LUC* (Col-0) or *CAB2::LUC* (Ws-2) were examined under free-running conditions in the presence or absence of 3% w/v sucrose with either monochromatic blue light (BL) or red light (RL) [14]. In both backgrounds, the free-running period (FRP) of *GI::LUC* decreased when sucrose was applied as previously reported (Figure 1 and Appendix A) [10,25]. Similarly, *CAB2::LUC* (assayed only under BL) also had an accelerated period, but this was limited to only the first time window of free-running conditions. In subsequent time windows, *CAB2::LUC* FRP returned to ~26 h, highlighting a temporal effect of sucrose on this reporter gene (Figure 2, Appendix A).

Alongside accelerating the circadian period, we also observed that the presence of sucrose acted to stabilize the circadian period and the accuracy of these rhythms under free-running conditions. In the absence of sucrose, the FRP of *GI::LUC* (Col-0) and *CAB2::LUC* (Ws-2) gradually accelerated over successive free-running time windows, and there was an increase in the relative amplitude error (RAE), which measures the accuracy of circadian oscillations (Figure 1 and Figure 2, Appendix A). This observed effect occurred independently of the prevailing light conditions. By contrast, seedlings treated with sucrose retained a consistently accelerated period, and RAE either stayed stable or decreased (rhythms became more accurate) for both *GI::LUC* (Col-0) or *CAB2::LUC* (Ws-2) (Figure 1 and Figure 2, Appendix A). Therefore, sucrose accelerates FRP and stabilizes circadian rhythms under free-running conditions, although there is a reporter gene effect in some instances.

### 3.2. The Oscillator Dynamically Responds to Sucrose under Free-Running Conditions

To determine if the observed effects of sucrose was caused by exposure when entrained or if the oscillator was dynamically responding to sucrose availability, plants were either entrained on media containing 0% w/v sucrose or 3% w/v sucrose and then moved to free-running conditions with media either containing 0% or 3% w/v sucrose. All experiments were carried under BL as BL results in the most stable rhythms under free-running conditions [40]. When sucrose was applied only during free-running conditions, there was a shortening of *CAB2::LUC* FRP (Figure 2A,B). However, in seedlings only exposed to sucrose during free-running conditions, FRP shortened more than when seedlings were exposed to sucrose during both entrainment and free-running conditions. Furthermore, seedlings only exposed to sucrose during free-running conditions did not display the temporal effect of sucrose on *CAB2::LUC* FRP (Figure 2A,B). Conversely, seedlings exposed to sucrose only during entrainment had a longer FRP than seedlings exposed to sucrose during free-running conditions.

As with the effect of sucrose on FRP, the ability of sucrose to stabilize circadian oscillations over successive free-running windows was also dependent upon sucrose being present during free-running conditions (Figure 2A,B). Prior exposure during entrainment was not sufficient for sucrose to stabilize circadian oscillations. Similar results were also obtained for *GI::LUC* (Col-0) for both the effect of sucrose on accelerating FRP and stabilizing the oscillator (Figure 1C,E, Appendix A). However, we did observe that *GI::LUC* seedlings exposed to sucrose only during entrainment had a quicker FRP than seedlings never exposed to sucrose. This effect was temporal and restricted to the first-time window of the experiment. In later time windows, the period of the two sets of seedlings became similar (Figure 1C, Appendix A). This would suggest that sucrose is dynamically regulating both periodicity and stabilizing FRP periodicity but may have distinct short- and long-term effects as highlighted previously [25].

### 3.3. GI Has a Light Dependent Sucrose Phenotype

Sucrose has been recently shown to signal directly to the oscillator by stabilizing GI to maintain robust oscillations during the dark [34]. To determine if *GI* was also involved in any light-dependent responses of sucrose, the response of *gi-11* to sucrose was examined under constant BL or RL. As described previously, *gi-11 GI::LUC* (assayed under RL) or *gi-11 CCA1::LUC* (assayed under BL) with or without exposure to sucrose had weaker circadian rhythms when compared to wild type, although sucrose did increase the amplitude and robustness of these rhythms for a limited period of time (Figure 3A,B, Appendix A) [41]. Wild-type seedlings displayed a consistent response to sucrose regardless of the light conditions or reporter gene used as described earlier (Figure 3A,B). By contrast, *gi-11* had distinct response to sucrose depending on whether BL or RL was used during free-running conditions. For BL, *gi-11* seedlings grown in the presence of 3% w/v sucrose containing media had a longer circadian period than *gi-11* seedlings grown in the absence of sucrose (Figure 3C,D, Appendix A). By contrast, we found that *gi-11* under RL displayed a wild-type response to sucrose in both the acceleration of free-running period and the stabilization of circadian rhythms over successive time windows (Figure 3A,B). However, in the absence of sucrose, *gi-11 GI::LUC* FRP under RL did not accelerate over successive time windows but instead de-accelerated (Figure 3A,B, Appendix A). As wild type responded consistently to BL and RL regardless of the reporter gene used, it would suggest that GI has a blue-light dependent role in sucrose signaling to the oscillator.

### 3.4. CCA1 and LHY Have Functionally Distinct Roles in Sucrose Signaling to the Oscillator

CCA1/LHY have been previously linked to sucrose signaling indirectly through a PIF metabolic-entrainment pathway, and separately through sugar-dependent regulation of PRR7 [10,32]. To determine if CCA1/LHY could have a direct role in sugar signaling, the response of *cca1-11*, *lhy-21*, and *cca1-11/lhy-21* mutants (all harboring *CAB2::LUC*) to sucrose was examined. The FRP of the respective mutants was firstly examined under RL across two consecutive free-running windows with varying concentrations of sucrose. As before, sucrose accelerated the FRP of *CAB2::LUC* (Figure 4A,B). This quickening of FRP occurred at even the lowest tested percentage of sucrose (0.5%), and there was an additive effect of increasing the percentage of sucrose on shortening FRP (Figure 4A,B). However, at higher percentages of sucrose, there was a decline in the magnitude of effect sucrose caused on FRP. Across all percentages of sucrose, the FRP remained stable across the two time windows, while the FRP of non-treated seedlings decreased as observed previously for *CAB2::LUC* (Figure 2 and Figure 4A,B). As with wild-type plants, sucrose decreased the period of *lhy-21*, and this occurred in a sucrose percentage dependent manner. By contrast, *cca1-11* was insensitive to sucrose application; across all tested concentrations of sucrose, there was no decrease in FRP for the first-time window and there was only a slight decrease at 2% sucrose in the second-time window. In the presence of sucrose, *cca1-11/lhy-21* mutants only had detectable oscillations for the 1st time window of free-running conditions before becoming arrhythmic as previously described [15]. During the first-time window, the FRP of *cca1-11/lhy-21* responded to the sucrose gradient as was seen in wild type and *lhy-21*, but surprisingly, at 3% sucrose, the FRP increased (Figure 4A,B).

As GI had a distinct response to sucrose depending on whether BL or RL was used, we looked to see if *cca1-11* or *lhy-21* also responded to sucrose differently depending on the light condition used. Plants were either entrained with or without 3% w/v sucrose and then transferred to free-running conditions (constant BL) with or without 3% w/v sucrose. As for RL, *cca1-11* seedlings exposed to BL were insensitive to sucrose application during the first-time window regardless of when plants were first exposed to sucrose (Figure 4C,D). However, during the second time window, *cca1-11* responded similarly to wild-type; seedlings exposed to sucrose during entrainment and under free-running conditions had an increase in *CAB2::LUC* FRP, while seedlings only exposed to sucrose during free-running conditions had no change in *CAB2::LUC* FRP when compared to the first time window (Figure 4C,D). As was seen for RL, sucrose supplied under free-running conditions shortened *lhy-21* FRP, but *lhy-21* was hypersensitive to the period shortening effects of sucrose compared to wild type. *lhy-21* seedlings exposed to sucrose during entrainment but not under free-running conditions also had a much shorter period compared to *lhy-21* seedlings never exposed to sucrose, as was seen previously for wild-type *GI::LUC* (Figure 1 and Figure 4A,B). Such a response was not seen in wild-type *CAB2::LUC* seedlings (Figure 2B and Figure 4D). By the second time window, this response was no longer observed, as was the case for *GI::LUC* before (Figure 1 and Figure 4D). CCA1 and LHY therefore seem to have functionally independent roles to sucrose signaling to the oscillator.

## 4. Discussion

It has become established that sucrose has a critical role in regulating the period and phase of the Arabidopsis oscillator. We have confirmed that sucrose application shortens the circadian period and also shown that the magnitude of this effect is dependent on the reporter gene analyzed. For *GI::LUC*, sucrose “permanently” shortened FRP across the length of the experiment, and this occurs regardless of the Arabidopsis background used (Figure 1, Appendix A). However, for *CAB2::LUC*, when sucrose is supplied during entrainment and under free-running conditions, FRP is only shortened during the first time window. This effect is not seen when sucrose is supplied exclusively during free-running conditions with FRP becoming consistently accelerated across the experiment (Figure 2, Appendix A), highlighting a short- and long-term response of *CAB2::LUC* to sucrose. The mechanism regulating these distinct processes is still not fully clear [25].

We have also shown that sucrose acts to stabilize the circadian oscillator across consecutive windows of free-running conditions (Figure 1, Figure 2 and Figure 4, Appendix A). Regardless of the reporter gene used, FRP gradually decreased if sucrose was absent under free-running conditions and RAE increased. Exposure to sucrose during entrainment was not necessary for the buffering effect to be observed, suggesting that sucrose is dynamically stabilizing the oscillator rather than fixing the oscillator in a stable state during entrainment. It has been previously shown that sucrose stabilizes circadian rhythms during the dark, and this was dependent on *GI* [25]. However, our experiments were carried out under BL or RL, and we did not observe a requirement for GI. In fact, GI under RL promotes the gradual acceleration of FRP (Figure 4B, Appendix A). This suggests that sucrose acts to stabilize the oscillator both in the light and in the dark, and this effect is regulated through two separate pathways.

It has been recently proposed that the Arabidopsis oscillator is not fixed but dynamically responds to both external and internal stimuli [23,42]. Our work here further supports such a fluid nature. When no sucrose was supplied during entrainment but then applied during free-running conditions, FRP shortened to the same extent as plants that have been entrained with sucrose (Figure 1, Figure 2, Figure 3 and Figure 4, Appendix A). Conversely, if sucrose was applied during entrainment but then removed under free-running conditions, FRP was not shortened and remained similar to plants never exposed to sucrose. However, in some instances, there seemed a prior memory of exposure of sucrose application (Figure 1 and Figure 4). The memory of sucrose was temporal and by the second time window, there was no or little difference in the FRP between the two treatment groups. This effect was also dependent on the reporter gene, highlighting again short- and long-term responses of circadian genes to sucrose availability. The oscillator also shows high sensitivity to sucrose, with concentrations as low as 0.5% w/v being sufficient to not only accelerate FRP but also to stabilize the circadian period under free-running conditions. Therefore, the effect of sucrose on the oscillator is not fixed but instead remains highly responsive to even low quantities of sucrose as availability fluctuates.

The pathways through which sucrose signals to the oscillator have begun to be uncovered. The sugar-responsive transcription factor bZIP63 directly regulates *PRR7*, sucrose stabilizes GI in the dark, and AKIN10 and TIC form an input pathway to regulate the phase of *GI* expression [26,28,34]. CCA1/LHY have been previously linked to sucrose signaling indirectly, either as a target of a PIF4 metabolic signaling pathway, or separately as a target of the SnRK1/bZIP63/PRR7 signaling module [32,34]. Here we found that CCA1 and LHY have direct but potentially independent roles in sucrose signaling to the oscillator. For both proteins, the effects of sucrose are dependent on the light condition. Under RL, *cca1-11* mutants are insensitive to sucrose-mediated shortening of FRP at most percentages across multiple time windows (Figure 4A,B). By contrast, under BL, *cca1-11* is insensitive to the effects of sucrose during the first time window but does respond to sucrose during the second time window in a similar manner to wild type (Figure 4C,D). This would therefore suggest that CCA1 forms part of a long-term, red-light dependent sucrose-signaling pathway but for BL is only part of a short-term signaling pathway. By contrast, LHY acts to restrict the effects of sucrose on the oscillator under BL across multiple time windows, indicating that LHY acts in a long-term signaling pathway. Additionally, LHY may also contribute to a short-term pathway, as *lhy* mutants have a much shorter period when exposed to sucrose only during entrainment compared to seedlings never exposed to sucrose (Figure 4). However, *lhy* mutants displayed a wild-type response under RL. CCA1 and LHY have conventionally been viewed as a singular component within the Arabidopsis clock, but independence has been highlighted in both temperature compensation and entrainment of the oscillator [38,43]. However, it remains unclear how these two highly similar proteins could be acting separately.

## Figures and Tables

**Figure 1 genes-10-00334-f001:**
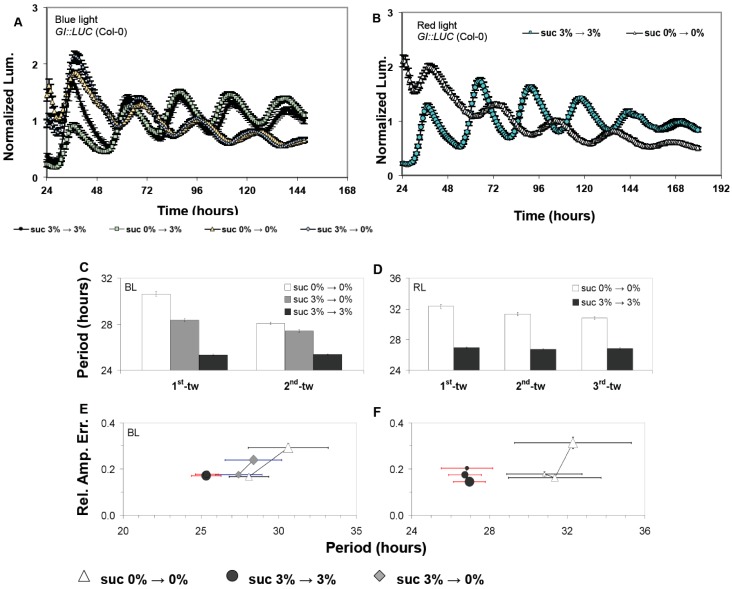
The effect of sucrose on rhythmic expression of the luciferase marker *GI::LUC*. The free-running period of *GI::LUC* seedlings (Col-0) under either monochromatic blue light (BL) (**A**,**C**,**E**) or red light (RL) (**B**,**D**,**F**). Panels (A,B) shows the time-course of average luminescence from representative experiments, while panels (C–F) show combined data from the independent experiments conducted as described above (six experiments under BL and three under RL). Under RL fast Fourier transform (FFT) analysis was performed during the time windows 06–101 h (1st-tw), 31–126 h (2nd-tw), and 56–151 h (3rd-tw) (beginning of free run was initiated at 0 h, as indicated in the figures). Under BL, FFT analysis was performed during the time windows 06–96 h (1st-tw) and 31–121 h (2nd-tw). Error bars represent standard error (SE). In (E) BL and (F) RL, the relative amplitude error (RAE) and circadian period are paired for all plants that generated an FFT output (RAE < 0.9; see Methods section); in these graphs, the closer a population is to the origin, the more robustly it oscillates.

**Figure 2 genes-10-00334-f002:**
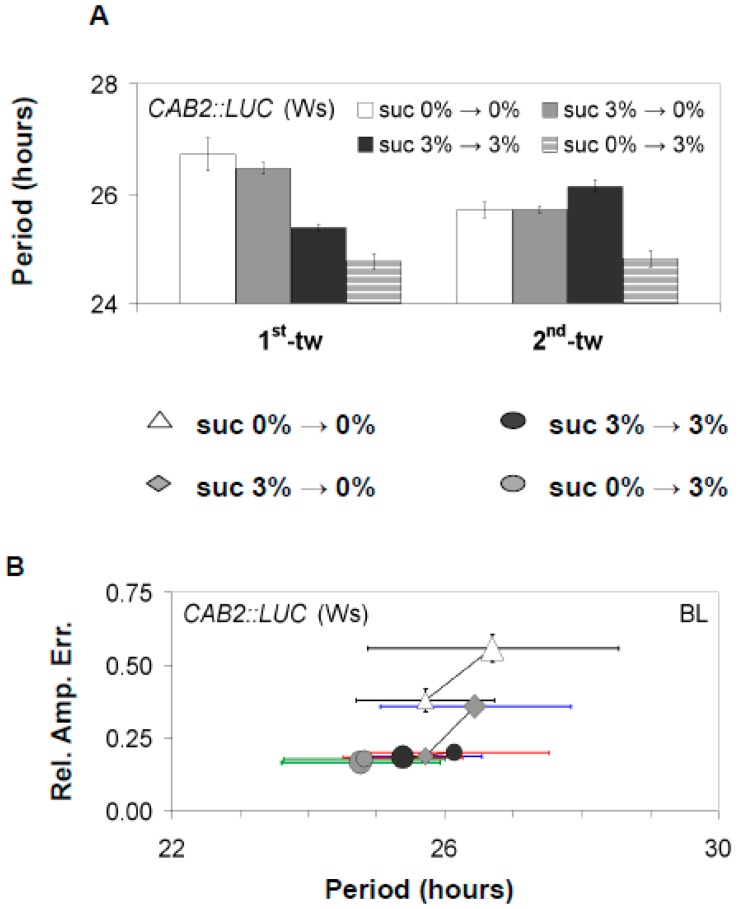
The effect of sucrose on rhythmic expression of the luciferase marker *CAB2::LUC*. C*AB2::LUC* (Ws-2) seedlings were either entrained in the presence or absence of 3% w/v sucrose and then moved to media with or without 3% w/v sucrose during free-running conditions (constant BL). (**A**) Analysis of periodicity in successive time windows of Ws-2 plants under different entrainment and free-running sucrose percentages. (**B**) As in Figure 1, the RAE and circadian period are paired for all plants that generated an FFT output (RAE < 0.9) under the different sucrose availabilities during entrainment or under free-running conditions. Error bars represent SE. FFT analysis was conducted as described in Figure 1 for the respective light conditions.

**Figure 3 genes-10-00334-f003:**
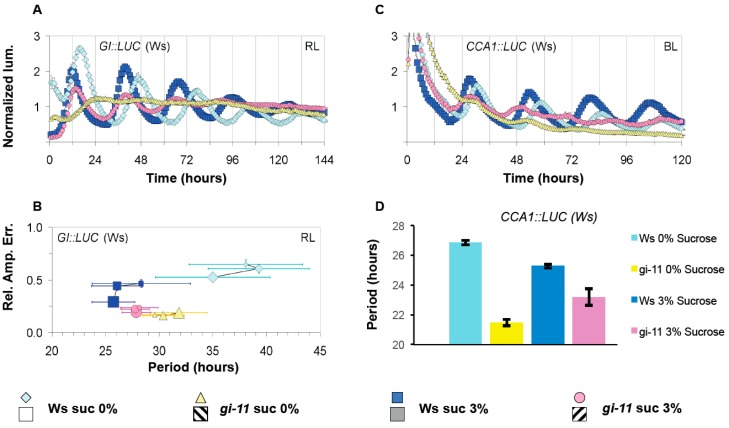
The effect of sucrose on oscillations in the *gi-11* mutant. The effect of sucrose on *gi-11 GI::LUC* (**A**) or *gi-11 CCA1::LUC* (**C**) oscillations under constant red light or constant blue light, respectively. (**B**) The paired RAE and circadian period of all plants under the different sucrose regimes, calculated as described in Figure 1 and Figure 2. (**D**) Periodicity estimates of (C). Error bars represent SE. Two independent experiments were performed for each respective light condition. Representative experiments are shown in (A,C), while panels (B,D) show combined data from the independent experiments.

**Figure 4 genes-10-00334-f004:**
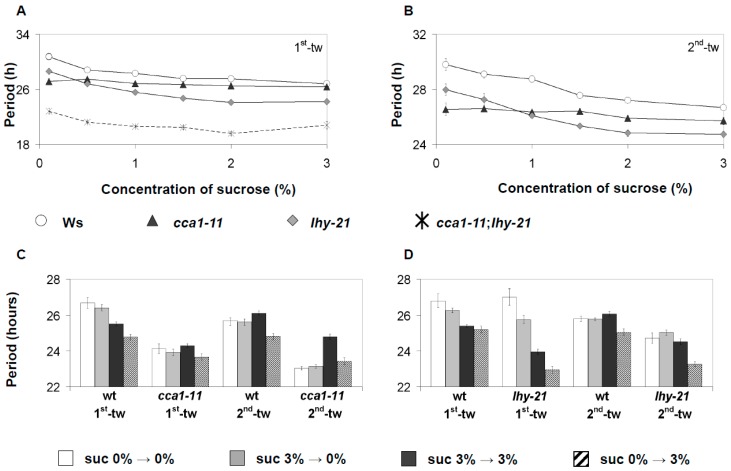
The effect of sucrose on oscillations of the *CAB2::LUC* marker in *cca1-11*, *lhy-21*, and *cca1-11/lhy-21*. (**A**,**B**) The free-running period of *cca1-11*, *lhy-21*, or *cca1-11/lhy-21* mutant harboring *CAB2::LUC* was examined across a sucrose gradient in two consecutive time windows under constant RL. Data were analyzed as described in Figure 1. (**C**,**D**) The free-running period of *cca1-11* (C) or *lhy-21* (D) under constant blue light was examined across two consecutive time windows. Seedlings were entrained with or no 3% w/v sucrose and then moved to media containing no or 3% w/v sucrose during free-running conditions. Data were analyzed as described in Figure 1. Error bars for (A–D) are standard error. Data represent the average of two independent experiments for both respective light conditions.

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
