# Peer review of "Physiological and Genetic Dissection of Sucrose Inputs to the Arabidopsis thaliana Circadian System"

_genes, 2019, doi:10.3390/genes10050334_

Reviewer 1 Report

Recently, sugar has been found to be a novel internal circadian entrainment signal. In this manuscript, the authors validated that sugars can speed the circadian period, however, is in a report gene dependent way. Further, they found the effect of sugar on circadian period and robustness relying on its availability in free running conditions. In terms of circadian robustness, they found sugar can enhance the robustness either in RL or in BL, as in dark reported previously. Interestingly, they characterized that CCA1 is required for short or long term of circadian responde to sugar, but LHY attenuates the effect of sugar on circadian period, suggesting their funcation is different in response to sugar of circadian clock. Overall, the well writen manuscipt is important for circadian field. I have a minor concern about the quality of figure. It seems that the descrepency between WT and gi-11 in figure 4 is not distinguishable. Similar problem in Fig 5C and D.

Another minor concern is the symbol for celsius degree in sencond section of method part. They should unify the form.

Author Response

Recently, sugar has been found to be a novel internal circadian entrainment signal. In this manuscript, the authors validated that sugars can speed the circadian period, however, is in a report gene dependent way. Further, they found the effect of sugar on circadian period and robustness relying on its availability in free running conditions. In terms of circadian robustness, they found sugar can enhance the robustness either in RL or in BL, as in dark reported previously. Interestingly, they characterized that CCA1 is required for short or long term of circadian responde to sugar, but LHY attenuates the effect of sugar on circadian period, suggesting their funcation is different in response to sugar of circadian clock. Overall, the well writen manuscipt is important for circadian field. I have a minor concern about the quality of figure. It seems that the descrepency between WT and gi-11 in figure 4 is not distinguishable. Similar problem in Fig 5C and D. 

Another minor concern is the symbol for celsius degree in sencond section of method part. They should unify the form. 

What was figure 4 (now figure 3) has been supplemented with an extra figure (supplementary figure 3) to further illustrate the effect of sucrose in the gi-11background. Comment on Celsius has been corrected.

Reviewer 2 Report

The authors have show in the present manuscript data supporting the importance of sucrose in the synchronization of the circadian clock in Arabidopsis thaliana. There main findings turn around the sensitivity of the clock's pace to very lo doses of sucrose and the preliminary characterization of the molecular pathway behind sucrose involvement in the frequency of the clock's periodicity and the robustness of it's oscillation. However because of flaws in data representation and lack of experiments in different conditions, the manuscript can not be accepted in its present version. The following comments will help improve the detected flaws and will make the manuscript ready for acceptance.

1. In Figure 1, data shown on panel A does not correspond to the legend, suc0%->3% is missing. It is true that later on this data is referenced on the manuscript but taking in consideration the argument exposed and to make things clear I will suggest to remove the oscillation of GI::LUC (Col-0) 0%->3% from this panel.

In panels A, C and E the oscillation, period and RAE of GI::LUC (Col-0) in suc3%->0% is shown.This shows nicely how addition of sucrose during entrainment only temporarily accelerates the pace of the clock. However this information is missing from panels B, D and F. I suggest to add these experimental results in order to prove the point that sucrose accelerates the pace of the clock and stabilizes the robustness of the circadian oscillations independently of the light conditions.

For panels A and B I suggest to reduce the size of the data points (triangle, circles, etc.) to make the results more clear.

2. In Figure 2 the control, suc0%->0%, is missing for panels A, C and E. If the statement that sucrose accelerates the pace of the clock in GI::LUC (WS) independently of the Arabidopsis accession, then this information is necessary in order to compare it to the results shown for Col-0 . Also to compare it to the results in suc3%->0% for WS. These should be the same or very similar to the ones in suc0%->0%.

For panels B, D and E the results for the oscillation, period and RAE in suc3%->0% is needed in order to support the argument here made (see point 1).

3. Figure 3 seems a bit confusing and should be redone freely by the authors but taking in consideration the following points.

- For panels C and D the control suc0%->0% is missing. This control is necessary to prove the argument that sugar doesn't changes the clock's periodicity under blue light and slightly under red light when using TOC1::LUC as reporter.

- CAB2::LUC reporter is used in suc0%->3% conditions to prove that acceleration of the clock's periodicity is possible by only adding sugar under free-running conditions. This is later compared to results in GI::LUC briefly showed in Figure 1A. Authors are strongly encouraged to add data on GI::LUC (WS)  (data in Col-0 is encouraged to be added in supplementary information) under suc0%->3% in this panel. It should include at least two of the following three information: GI::LUC oscillation, period or RAE.

The phrases between lines 175 and 183 should be carefully re-written to make the point stated more clear.

4. In Figure 4 panel D, the background color for the histograms corresponding to gi-11 suc 0% and gi-11 suc 3% should be changed to match their corresponding legend. 

 It is also be a good idea to add the information relative to period and RAE for both reporters used (GI::LUC and CCA1::LUC). This in order to make more clearer the differences in the response of the gi-11 mutant under different light conditions. If the authors find that these additional data could lead to confusion, I would suggest to add it in supplementary information.

5. In Figure 5, the legend for the axis of panels A and B are missing (Period (hours) in y-axis and concentration of sucrose on x-axis). Also for al panels the authors should make clear in each graph the light condition under which the experiments were performed, red light or blue light.

On panels C and D, the background color for the histograms corresponding to suc 0% -> 3%  should be changed to match their corresponding legend. 

On line 239 it is said that cca1-11/lhy-21 does not show oscillation during the second time window analysed and that this data is not shown. If the authors decide not to present the data this phrase should be removed they should reference previous studies showing the arryhtmicity of the cca1-11/ly-21 double mutant.

Overall the authors should be extra careful when referencing the figures in the text. Some of them do not coincide which the results being described.

Author Response

In Figure 1, data shown on panel A does not correspond to the legend, suc0%->3% is missing. It is true that later on this data is referenced on the manuscript but taking in consideration the argument exposed and to make things clear I will suggest to remove the oscillation of GI::LUC (Col-0) 0%->3% from this panel.

The missing data has been re-added to the figure. 

In panels A, C and E the oscillation, period and RAE of GI::LUC (Col-0) in suc3%->0% is shown.This shows nicely how addition of sucrose during entrainment only temporarily accelerates the pace of the clock. However this information is missing from panels B, D and F. I suggest to add these experimental results in order to prove the point that sucrose accelerates the pace of the clock and stabilizes the robustness of the circadian oscillations independently of the light conditions.

This data has been moved to supplementary figure 2 – was meant to show that the effects of sucrose under free-running conditions is not restricted to a particular ecotype.

For panels A and B I suggest to reduce the size of the data points (triangle, circles, etc.) to make the results more clear.

This suggestion as enacted.

2. In Figure 2 the control, suc0%->0%, is missing for panels A, C and E. If the statement that sucrose accelerates the pace of the clock in GI::LUC (WS) independently of the Arabidopsis accession, then this information is necessary in order to compare it to the results shown for Col-0 . Also to compare it to the results in suc3%->0% for WS. These should be the same or very similar to the ones in suc0%->0%.

For panels B, D and E the results for the oscillation, period and RAE in suc3%->0% is needed in order to support the argument here made (see point 1).

Figure 2 was modified as a response.

3. Figure 3 seems a bit confusing and should be redone freely by the authors but taking in consideration the following points.

Figure 3 has been re-worked, with TOC1::LUC moved to supplementary material and extra data added (relative amplitude error plots), so figure 2 (what was figure 3) has become only focused on CAB2::LUC

- For panels C and D the control suc0%->0% is missing. This control is necessary to prove the argument that sugar doesn't changes the clock's periodicity under blue light and slightly under red light when using TOC1::LUC as reporter.

- CAB2::LUC reporter is used in suc0%->3% conditions to prove that acceleration of the clock's periodicity is possible by only adding sugar under free-running conditions. This is later compared to results in GI::LUC briefly showed in Figure 1A. Authors are strongly encouraged to add data on GI::LUC (WS)  (data in Col-0 is encouraged to be added in supplementary information) under suc0%->3% in this panel. It should include at least two of the following three information: GI::LUC oscillation, period or RAE.

Figure 3 was modified as a response.

The phrases between lines 175 and 183 should be carefully re-written to make the point stated more clear.

Extra text has been added (lines 171-176)

4. In Figure 4 panel D, the background color for the histograms corresponding to gi-11 suc 0% and gi-11 suc 3% should be changed to match their corresponding legend.

Fixed

 It is also be a good idea to add the information relative to period and RAE for both reporters used (GI::LUC and CCA1::LUC). This in order to make more clearer the differences in the response of the gi-11 mutant under different light conditions. If the authors find that these additional data could lead to confusion, I would suggest to add it in supplementary information.

As suggested by the reviewer, an extra figure has been added to the supplementary material with the requested data (Supplementary figure 3)

5. In Figure 5, the legend for the axis of panels A and B are missing (Period (hours) in y-axis and concentration of sucrose on x-axis). Also for al panels the authors should make clear in each graph the light condition under which the experiments were performed, red light or blue light.

Fixed

On panels C and D, the background color for the histograms corresponding to suc 0% -> 3%  should be changed to match their corresponding legend. 

All legend colors match.

On line 239 it is said that cca1-11/lhy-21 does not show oscillation during the second time window analysed and that this data is not shown. If the authors decide not to present the data this phrase should be removed they should reference previous studies showing the arryhtmicity of the cca1-11/ly-21 double mutant.

A new reference has been added and extra data has also been added (supplementary figure 4). 

Reviewer 3 Report

Comments on philippou et, al., 2019

In their MS, the authors investigated the role of sucrose with respect to circadian clock. Since role of photosynthate has been well studied in past the current findings adds to the further knowledge of sucrose in clock. The manuscript is well written but at the same time a bit complicated to understand the results.

However, authors need to address some of the points stated below that will help clarify the results in the MS.

1.      I was wondering if authors could explain in brief the rationale behind using BL and RL conditions so that readers don’t have to look for references. Also, I can see some differences in luminescence in GI:LUC in Col and Ws backgrounds. I was wondering if the difference in luminescence is due to a natural phytochrome mutation in Ws ecotype. Authors need to address, why these ecotypes were selected for analyzing interactions as they can have different levels of metabolic signatures.

2.      In Figure 1, 2 and 3 it would be better if authors can represent the graphs and luminescence data in color. I can also see 4 luminescence values in Figure1A and only 2 luminescence values in Figure 1B are the authors missing some treatment of sucrose?

3.      On line 120 page 3 the authors talk about RAE which does not corresponds to figure 1C-D, 2C-D, Figure 3 I think it should have been Figure1E-F, 2E-F and Figure3B

4.      It would also be good to see the luminescence graph for CAB2:LUC along with TOC1:LUC in figure3 in BL and RL. At the same time it would also be interesting to see if authors can compare the results of CAB2:LUC and TOC1:LUC in Ws background to that of Col-0 background

5.      At page2 line 48 reference no 14 is not in context with the paragraph

6.      I feel that physiologically 3% sucrose is too higher concentration and it would be nice to see a preliminary data with different % of sucrose to know that if it’s not a biphasic response of sucrose.

Author Response

1.      I was wondering if authors could explain in brief the rationale behind using BL and RL conditions so that readers don’t have to look for references. Also, I can see some differences in luminescence in GI:LUC in Col and Ws backgrounds. I was wondering if the difference in luminescence is due to a natural phytochrome mutation in Ws ecotype. Authors need to address, why these ecotypes were selected for analyzing interactions as they can have different levels of metabolic signatures.

Col and Ws-2 are the major accessions that have ben used in clock research. GI differences could reflect differing transgene insertion sites.

2.      In Figure 1, 2 and 3 it would be better if authors can represent the graphs and luminescence data in color. I can also see 4 luminescence values in Figure1A and only 2 luminescence values in Figure 1B pare the authors missing some treatment of sucrose?

 Figure 1 has been colorized as requested by reviewer 2. 

3.      On line 120 page 3 the authors talk about RAE which does not corresponds to figure 1C-D, 2C-D, Figure 3 I think it should have been Figure1E-F, 2E-F and Figure3B

Fixed

4.      It would also be good to see the luminescence graph for CAB2:LUC along with TOC1:LUC in figure3 in BL and RL. At the same time it would also be interesting to see if authors can compare the results of CAB2:LUC and TOC1:LUC in Ws background to that of Col-0 background

Raw RAE plots have been added from figure 1, figure 2 and supplementary figure 1 to make supplementary figure 2. This allows you to compare the effects of sucrose on periodicity and accuracy across successive time windows across genotypes and reporter genes.

5.      At page2 line 48 reference no 14 is not in context with the paragraph

Fixed

6.      I feel that physiologically 3% sucrose is too higher concentration and it would be nice to see a preliminary data with different % of sucrose to know that if it’s not a biphasic response of sucrose.

Figure 4 has a range of sucrose % and at all test percentages there was a decrease in periodicity. 

Round  2

Reviewer 2 Report

The authors have performed many of the previously suggested changes on the manuscript. However some comments have not been addressed or no explanation of why has been given.

For panels B, D and F (supplementary figure 1) the results for the oscillation, period and RAE in suc3%->0% is needed in order to support the argument here made (see point 1).

For panels C and D (supplementary figure 3) the control suc0%->0% is missing. This control is necessary to prove the argument that sugar doesn't changes the clock's periodicity under blue light and slightly under red light when using TOC1::LUC as reporter.

In Figure 2 CAB2::LUC reporter is used in suc0%->3% conditions to prove that acceleration of the clock's periodicity is possible by only adding sugar under free-running conditions. This is later compared to results in GI::LUC briefly showed in Figure 1A. Authors are strongly encouraged to add data on GI::LUC (WS)  (data in Col-0 is encouraged to be added in supplementary information) under suc0%->3% in this panel. It should include at least two of the following three information: GI::LUC oscillation, period or RAE.

Author Response

Please find attached a revised version of our manuscript entitled "Physiological and genetic dissection of sucrose inputs to the Arabidopsis thaliana circadian system".We have removed supplementary figures 1, 3, and 5, which the reviewers had concerns with. The supplementary figures 2 and 4 have now be re-named 1 and 2 respectively and the text and references has been updated accordingly to match. The removal of these figures does not change or affect any conclusions that were made within the manuscript. We believe we have now fully addressed all reviewers’ comments. The manuscript has been much improved by these suggested changes and would like to thank the reviewers for all the helpful comments. We look forward to hearing your views on the revised version.

Reviewer 3 Report

The Authors have incorporated changes but still there are some minor issues

please check all the figure legends carefully as i can see on line 191 page 6 two symbols of different shapes, which are not required

it is confusing for me why authors have not used same set of treatment in Figure 4 and supplementary figure 5.

Also, it is surprising for me that in Supplementary figure 5 in absence of sucrose the rhythms in cca1-11 mutant is dampened but not in cca1-11;lhy-21 it still exists. can the authors explain it well.

in figure 4 B the data for cca1-11lhy-21 mutant is missing in 2nd TW which i suppose to be there if it still oscillates as seen in supplementary fig 5.

Author Response

(The authors gave the same response as above.)
